# Genetic Variability and Population Structure in the Sardinian Anglo-Arab Horse

**DOI:** 10.3390/ani10061018

**Published:** 2020-06-11

**Authors:** Andrea Giontella, Francesca Maria Sarti, Irene Cardinali, Samira Giovannini, Raffaele Cherchi, Hovirag Lancioni, Maurizio Silvestrelli, Camillo Pieramati

**Affiliations:** 1Department of Veterinary Medicine—Sportive Horse Research Center, University of Perugia, via S. Costanzo 4, 06123 Perugia, Italy; maurizio.silvestrelli@unipg.it (M.S.); camillo.pieramati@unipg.it (C.P.); 2Department of Agricultural, Food and Environmental Sciences, University of Perugia, Borgo XX Giugno, 74, 06121 Perugia, Italy; francesca.sarti@unipg.it (F.M.S.); samira.giovannini@gmail.com (S.G.); 3Department of Chemistry, Biology and Biotechnology, University of Perugia, via Elce di Sotto, 8, 06123 Perugia, Italy; irene.cardinali@unipg.it (I.C.); hovirag.lancioni@unipg.it (H.L.); 4AGRIS, Servizio Ricerca Qualità e Valorizzazione delle Produzioni Equine, piazza D. Borgia, 4, 07014 Ozieri, Sassari, Italy; ilex283@gmail.com

**Keywords:** Sardinian Anglo-Arab horse, population structure, genetic variability, mitochondrial DNA

## Abstract

**Simple Summary:**

It is known that animal populations might be affected by bottleneck phenomena, which reduce genetic variability, increase inbreeding and consequentially reduce evolutionary potential. Pedigree completeness, genetic variability and population structure were analyzed in the Sardinian Anglo-Arab horse (SAA) breed, and the analyzed results were compared with three other Italian horse populations: Maremmano, Murgese and Bardigiano (reported in bibliography). In addition, the maternal lineage was analyzed through mitochondrial DNA in order to preserve and improve the breed. The estimated parameters suggest that the SAA breed is well managed, especially when considering the female lineage because it still conserves a high number of founder mares. The role of breeding programs in the conservation of genetic diversity is essential. In fact, a breeding program might lead to fast genetic progress; however, it might also lead to a high level of inbreeding and consequently to a genetic drift. Analyzing both these parameters and the additional use of mitochondrial DNA might be an effective tool not only to verify the success of a breeding program, but also to be helpful for breeders on planning effective mating programs.

**Abstract:**

The Sardinian Anglo Arab (SAA) is a famous horse breed in Italy, with a significant historical background in the island of Sardinia. The aim of the study is to perform an evaluation of genetic variability in SAA using pedigree and mitochondrial data. In the current population, pedigree completeness was observed to be close to 100%, while the inbreeding coefficient and the average relatedness were lower than 3%. The ratio of effective founders/numbers of ancestors was 3.68 for the whole pedigree. The effective population size (N_e_) computed by an individual increase in inbreeding (N_e_1_) was 456.86, the N_e_ on equivalent generations (N_e_2_) was 184.75, and this value slightly increased to 209.31 when computed by log-regression on equivalent generations (N_e_3_). These results suggest the presence of crossbreeding and bottleneck phenomena, and they were compared with other Italian horses (reported in bibliography) to present the SAA among the Italian horse breeds scenario. Furthermore, the noteworthy mitochondrial variability reflects the use of a considerable number of founder mares; the contribution of L lineage was very important, probably because of the re-colonization from the Iberian Peninsula after the Last Glacial Maximum.

## 1. Introduction

The Sardinian Anglo-Arab (SAA) is a horse breed that is mainly used for equestrian sport. It originated by crossing indigenous mares with old Arabian lines (Purosangue Orientale) and Thoroughbred stallions, and the use of these two breeds was very different depending on the historical period. Arabian and Thoroughbred stallions were alternatively used in order to achieve different goals, such as racing, jumping and other equestrian disciplines [1]. This management strongly affected the breed composition and demographic and genetic patterns of the SAA [2].

In 1874, the Ozieri Army Remount Station was established to supply mounts for the cavalry units of the Italian Army. Indigenous Sardinian mares were crossed with oriental-bred stallions to produce resistant and fast horses that were suitable for the cavalry [3,4].

In the early 1900s, Captain Grattarola, the commander of the Ozieri Remount Station, continued the work by crossing the best available indigenous mares with Purosangue Orientale stallions which had been purchased directly from Bedouin tribes. Later on, Arab and Thoroughbred stallions started to be bred with the mares of mixed Sardinian and oriental ancestry [4].

In 1967, when the studbook of this breed was officially established, blood percentages were defined and accurately settled. The breed was officially named “Sardinian Anglo-Arab,” and it was described as, “The product of the selection and crossbreeding between Arabian lines, Thoroughbred, Anglo-Arab Thoroughbred, Sardinian Anglo-Arab stallions and indigenous mares with a percentage of Arabian blood not less than 25% and not more than 75%.” From that moment, the main breeding goal became sport [4,5].

SAA reference populations were analyzed and compared with three other Italian horse breeds, Bardigiano, Maremmano and Murgese, which were studied with the same statistical approach. This horse breed lives in the isolated geographic context of Sardinia (an Italian island in the Mediterranean Sea) which could have had a key role in shaping its population and genetic structure. Recent studies based on horse mitogenomes have defined a complex and detailed phylogeny, describing the mutational motifs of 18 different haplogroups [6] and the mitochondrial variability of different Italian local breeds [7,8]. However, data at a micro-geographic level for SAA breed are still lacking. In this context, the present study analyzed the SAA population to define its structure and genetic variability. By considering its crossbreeding origins, we aimed to shed light on the maternal lineage and to contribute to the preservation and the improvement of SAA.

## 2. Materials and Methods

### 2.1. Reference Populations

In this study, three reference populations (REF) were considered: population REF1 consists of all the 43,624 horses registered in the studbook kept by the Italian Ministry of Agricultural Policies; REF2 consists of 22,956 horses, including only Anglo-Arab, Sardinian Anglo-Arab, Thoroughbred, Arab and Sardinian Arab (14,305 females and 8651 males); REF3 includes only living horses, for a total of 13,266 animals (7141 females and 6125 males).

The results obtained from these populations were compared with those observed in the reference populations of the following three other Italian breeds: Bardigiano with one reference population (REF_B) [9], Maremmano with four reference populations (REF1_MA, REF2_MA, REF3_MA, REF4_MA) [10] and Murgese with one reference population (REF_M) [11], as reported in Table 1.

### 2.2. Pedigree Analysis and Genetic Variability

The Agricultural Research Agency of Sardinia—Istituto Incremento Ippico della Sardegna (AGRIS) provided a pedigree file that included 43,624 horses (27,052 females and 16,572 males) born between 1853 and 2013 which were registered in the studbook kept by the Italian Ministry of Agricultural Policies.

Several parameters related to pedigree completeness (number of fully traced generations—FTG, maximum number of generations traced—MNGT, equivalent complete generations—ECG), genetic variability (individual inbreeding—F, average relatedness coefficients—AR, effective number of ancestors—f_a_ only in REF1, genetic conservation index—GCI) and population structure (effective population size—N_e_, generation interval—GI) were computed using “ENDOG v 4.8” software [12], as reported in a previous study [10]. Regarding the inbreeding—F, a further analysis was performed to verify the statistical significance of the difference between SAA REF1 and REF3 with a Student’s *t*-test computed by the R Core Team [13]. To prevent a possible misinterpretation of the N_e_ value, because of the small number of individuals and the overlapping generations [14], it was computed in the following ways: N_e_1_ (individual increase in inbreeding), N_e_2_ (regression on the number of the equivalent generations) and N_e_3_ (logarithmic regression on the number of equivalent generations). The ratio N_e_1_/2 was also calculated because it is a parameter that can explain whether bottleneck phenomena occurs in the population. As reported in Giontella et al. [10], if the genetic drift has stabilized in a population, N_e_1_/2 should be close to the value of f_e_.

All the parameters were estimated in all the reference populations, while GI was computed only on REF1.

### 2.3. Mitochondrial DNA Variation

To provide a comprehensive overview of the SAA maternal genetic variability, we phylogenetically analyzed 56 mitochondrial DNA (mtDNA) control-region sequences (Appendix A) and compared them to 170 mitogenomes from the other three horse breeds (Appendix A): Bardigiano (*n* = 48), Maremmano (*n* = 90) and Murgese (*n* = 32). The mtDNA control region was amplified by PCR (Swift MaxPro, Esco; Hatboro, PA, USA) using forward 5′-AAACCAGAAAAGGGGGAAAA-3′ and reverse 5′-TGGCGAATAGCTTTGTTGTG-3′ oligonucleotides. PCR reactions contained 1× Buffer GoTaq (Promega Corporation; Madison, WI, USA), 2.5 mM of each dNTPs, 0.3 μM of each primer, 0.03 U/μL of GoTaq DNA polymerase (Promega Corporation; Madison, WI, USA), 30 ng of genomic DNA and H_2_O to a final volume of 25 μL. PCR amplification was carried out as follows: 95 °C for 2 min, followed by 35 cycles of 95 °C for 30 s, 55 °C for 30 s, 72 °C for 120 s and then 72 °C for 5 min. The PCR fragment was purified using exonuclease I and alkaline phosphatase (ExoSAP-IT enzymatic system—USB Corporation, Cleveland, OH, USA), and subsequently Sanger sequenced with the primer forward 5′-CACCCAAAGCTGAAATTCTA-3′. Sequences were aligned to the Equine Reference Sequence (ERS; NC001640; https://www.ncbi.nlm.nih.gov/nuccore/NC_001640) for the haplotype annotation, and their evolutionary relationships were evaluated through a median-joining tree built using “Network” software v.10.0” [15].

## 3. Results

### 3.1. Pedigree Completeness

The completeness of pedigree information is presented in Figure 1.

The plot represents the number of ancestral generations that can be traced.

From the first to the fifth generation, the REF1, containing the ancestor of the breed, showed a completeness of 86.44%, 79.87%, 75.82%, 72.88%, 68.53%; in the REF2 population these values increased up to 92.65%, 89.72%, 88.33%, 87.28%, 85.30%; as expected, in the REF3 population, the values reached 100%: 99.96%, 99.01%, 99.01%, 98.96%, 98.72%.

The mean of the maximum number of traced generations reported in the studbook (REF1) was equal to 11.15; the mean of the fully traced generations was 3.62; and the mean of the equivalent complete generations was 5.96. In the REF2, these values increased to 13.09, 7.30 and 4.00, respectively; in the living animals (REF3), these numbers increased to 15.40, 8.66 and 5.43.

### 3.2. Genetic Variability

The average inbreeding coefficient (F) in the REF1 population was 0.012 and its trend increased by year of birth up to a maximum of 0.023 in 2010 (Figure 2).

The F was 0.017 in REF2 and 0.021 in REF3. The difference between REF1 and REF3 F values was statistically significant (*p* ≤ 0.05) according to t-test. The average relationship coefficient (AR) in REF1 was 1.72% and it increased up to 2.48% in REF2 and to 2.83% in REF3. The effective numbers of founders (f_e_) of the investigated SAA populations were 287 in REF1, 190 in REF2 and 157 in REF3; nevertheless, only 316 individuals explained the 50% of the genetic variability in REF1, 26% in REF2 and 64% in REF3. The effective ancestors (f_a_) numbers were 78 in REF1, 48 in REF2 and 38 in REF3; the ratios of f_e_/f_a_ were 3.68 in REF1, 3.96 in REF2 and 4.13 in REF3.

In the REF1, the mean genetic conservation index (GCI) was 26.17 ± 19.58, with a minimum of 0.90, a maximum of 119.22 and a coefficient of variation of 70%.

### 3.3. Population Structure

The effective size of SAA population N_e_1_ was the following: 456.86 in REF1, 196.32 in REF2 and 187.02 in REF3; N_e_2_ was 184.75 in REF1, 127.86 in REF2 and 88.34 in REF3; N_e_3_ values in the three REF_SAA were, respectively, 209.31, 128.52 and 87.36.

The ratio N_e_1_/2 in REF1 was 228.43 and decreased to 98.16 and to 93.51 in REF2 and in REF3, respectively. The mean of the generation interval (GI) estimated on the whole pedigree (REF1) was 10.98 ± 5.10 years with a range of 8.42 ± 5.19 (mother–daughter) to 12.24 ± 5.01 (father–son).

### 3.4. Mitochondrial DNA Variation

The analysis of 610 base pairs of the 56 SAA control-region sequences showed a high haplotype diversity (Hd = 0.969), with a total of 31 haplotypes and 49 polymorphic sites detected (S).

All the SAA haplotypes were classified into eight haplogroups (A, B, E, G, I, L, M and N), with a prevalence of L haplogroup (41%), followed by G and I haplogroups (16% and 14%, respectively) (Figure 3).

Network analysis of all mitochondrial control regions belonging to SAA, Bardigiano, Maremmano and Murgese showed the phylogenetic relationships between haplotypes and the haplogroup clustering (Figure 4).

A total of 18 unique SAA haplotypes could be observed: one from haplogroup A, two from B, two from E, one from G, two from I, seven from L and three from M. Among the remaining haplotypes, six were shared only with Maremmano (two belonging to haplogroup G, one to I and three to L), while seven haplotypes were also shared with Bardigiano and/or Murgese breeds (Figure 4). The haplotype with the highest frequency among SAA samples was unique and belonged to haplogroup L.

## 4. Discussion

### 4.1. Pedigree Completeness

The structure and the pedigree of the SAA were compared with those of three other Italian breeds in order to underline possible differences and similarities (Table 1).

The comparison with the REF3_MA [10], which referred to all the Maremmano animals registered in the ANAM Studbook (1980), shows that both breeds have a good percentage of pedigree completeness; however, the following differences were observed: the REF2 had a lower percentage of pedigree completeness than REF3_MA (five generations) that reached 99.50%, 97.20%, 91.60%, 83.20%, 70.90%; in both populations of living animals (REF3 and REF4_MA) the pedigree completeness reached 100%.

The quality of a pedigree depends on its length and depth: the more a pedigree is complete in terms of the number of generations traced, the more it is possible to determine the real number of ancestors. Therefore, the degree of pedigree information can affect the average inbreeding coefficient. All parameters that describe the probability gene origin are affected by the pedigree depth; in fact, a suitable estimation of genetic variability widely depends on available and accessible pedigree information measured by pedigree completeness [16].

Regarding the mean of the maximum number of traced generations and the mean of fully traced generations (Table 2), REF3_MA, REF4_MA and REF_M showed lower values (10.50, 3.30; 11.60, 3.80; 8.84, 3.28) than the ones estimated in SAA REF1 and REF2. As for the mean of the equivalent complete generations, a different situation was observed: only REF4_MA showed the highest value equal to 6.40; this value confirms that it is the most complete and deep pedigree.

Considering the same parameters, the REF_M showed quite similar values (8.84, 3.28 and 5.00) to those of REF4_MA, and therefore, lower ones than those of SAA.

### 4.2. Genetic Variability

The average inbreeding coefficient in SAA was lower than those of the other breeds (Table 3).

The F in REF2 was 0.017; in REF3_MA it was 0.029 [10]; in REF_M it was 0.045 [11]; in REF_B the highest value was 0.08 [9]. According to the average inbreeding coefficient results, SAA reached similar values to Maremmano, while Murgese results are more comparable with the values reached by Bardigiano. These differences could be due to the different numbers of horses registered in the studbooks—22,956 and 15,875 respectively for REF3 and Maremmano, and 2366 and 9469 for Murgese [11] and Bardigiano [9].

A comparison between the AR coefficients was carried out in SAA reference populations and in the populations registered in the official studbooks of the other breeds. AR is a very useful parameter since it allows breeders to preserve the genetic pool. Horses with lowest AR, used as stallions and mares, can reduce the inbreeding, balancing the gene contributions of the founders in the population, and consequently the genetic variability. In REF2, the AR was lower than those in REF3_MA and REF4_MA, which were respectively 5.52% and 6.13%, although it was similar to the REF_M (2.45%); in the REF_B, this parameter increased to 11.0% [9,10,11]. The SAA results showed an increase in inbreeding (*p* ≤ 0.05) between all the horses registered in the studbook (REF1) and the living horses (REF3), which provides breeders and owners with important information because inbreeding control is fundamental for mating plans. It must be highlighted that the results for this parameter allow us to verify the efficiency of a breeding program. Nevertheless, SAA shows limited values for F and for AR which reveal the effort to control inbreeding through applied breeding strategies. This demonstrates that it is possible to maintain genetic diversity in SAA by mating animals with lower AR. Maremmano showed a rather low F value, but a higher AR value, which might reveal a possible lack of suitable mating in its breeding program. Conversely, Murgese showed the smallest population in terms of consistency, despite reaching values similar to the other breeds for both F and AR parameters. It is possible to observe that Murgese shows AR and F values lower than those of Bardigiano, which has a larger population, and the AR value is closer to that of the REF2 SAA. This was unexpected, so we can assume that better strategies have been applied to the Murgese breeding programs in order to avoid inbreeding.

The effective number of founders (f_e_), which represents the number of equally contributing founders that are expected to produce the same genetic diversity of the population, was higher in SAA than in Maremmano (74 in REF3_MA, 64 in REF4_MA). In addition, the effective number of ancestors (f_a_), which is the minimum number of ancestors, not necessarily founders, and which explains the whole genetic diversity of a population, was higher than those estimated in REF3_MA (30) and in REF4_MA (25) by Giontella et al. [10]. The ratio f_e_/f_a_ in SAA was higher than that observed in the Maremmano population (2.47 in REF3_MA, 2.56 in REF4_MA).

In REF_B, f_e_ was equal to 18 and f_a_ to 14 [9]; the ratio f_e_/f_a_ was 1.20; in the REF_M, f_e_ was 37, f_a_ was 20 and the ratio f_e_/f_a_ was equal to 1.85 [11]; even these results are lower than those in SAA reference populations.

The ratio f_e_/f_a_ is very important because it indicates if the analyzed population is affected by genetic drift, and therefore shows bottleneck (f_e_/f_a_ > 1) [10]. In this research, the reference populations of both SAA and Maremmano showed higher values compared with the breeds that had smaller numbers of individuals (Murgese and Bardigiano). According to these results, it might be assumed that suitable breeding programs were carried out in the Murgese and Bardigiano to preserve their genetic variability. On the other hand, there was a massive use of the same stallions in the SAA (approximately 50% of offspring are derived from only about 3% of sires) and Maremmano breeding programs; therefore, a bottleneck risk occurs in these two breeds. A further confirmation is provided by the number of ancestors that explains the 50% of genetic variability. Comparing the results in REF2 (26), REF3_MA (13), REF_M (7) and REF_B (8), the number of ancestors used for reproduction is higher in SAA than in the other breeds, which is explained by the breeding strategies applied for this breed. Breeders of SAA are permitted to choose stallions from the three following genetic types: Thoroughbred, Anglo Arab and Sardinian Anglo Arab. However, the stallions, which are available for reproduction, are probably not used in a suitable way as confirmed by the high f_e_/f_a_ ratio.

The GCI computes the genetic contributions of all the identified founders; for this reason, it has been assumed that the animals, which have a good number of genes percentage from a large number of founders, get higher values of GCI [17]. In the population REF1, the mean of GCI was 26.17 ± 19.58, with a minimum of 0.90 and a maximum of 119.22 and a variability coefficient of 70%. In REF3_MA, the GCI mean was 5.50 ± 3.37, ranging from a minimum of 0.81 to a maximum of 21.32; it was also correlated with a lower variability (61%) [10].

### 4.3. Population Structure

The population structure parameters in reference populations are reported in Table 4.

The effective population size N_e_ is a key parameter for planning strategies to define and protect endangered animals. In other words, a population with low N_e_ has a higher probability of extinction [16]. As reported in the Materials and Methods, ENDOG software computes this parameter in three different ways [12]. The effective population size N_e_1_ was equal to 68.10 in the REF3_MA and 71.20 in the REF4_MA; N_e_2_ was 42.00 in REF3_MA and 36.60 in REF4_MA; N_e_3_ was 42.20 and 36.90 in the same reference populations (Table 4). All these values are lower than those estimated on SAA, confirming that this population seems less endangered than the other studied breeds at this moment; unfortunately, these parameters were not available for Murgese and Bardigiano.

As previously reported, the ratio Ne__1_/2 in REF1 is 228.43, and it decreases to 98.16 and 93.51 in REF2 and in REF3. These values are distant from the effective number of founders of the three above mentioned reference populations (287, 190 and 157), thereby confirming that bottleneck phenomena may occur. The same scenario occurred in REF3_MA and REF4_MA (34 and 36 respectively), as referred to by Giontella et al. [10].

The SAA medium generation interval (GI) is 10.98 ± 5.10 years and it is similar to the GI in the Maremmano [10], which is 10.65 ± 4.72. The GI ranged between 9.81 ± 4.32 (mother–daughter) and between 10.40 ± 4.71 (father–daughter), and it is also similar to the Murgese [11] wherein the GI is equal to 11.7 ± 10.90; in the Bardigiano [9] this value decreases to 8.74 (mother–daughter 8.67, mother–son 9.18, father–daughter 8.45 and father–son 8.65). In each studied breed, apart from the Bardigiano, the GI calculated was longer because the SAA, Maremmano and Murgese horses are usually mated for a long period. In SAA, this result is attributable to the practice of using stallions and mares for a long period of time, even after their sporting careers.

### 4.4. Mitochondrial DNA Variation

The analysis of SAA mitochondrial control region sequences suggests a high level of haplotype diversity, which is comparable to other continental Italian breeds and essentially identical to those from Maremmano and Murgese; nevertheless, it is higher than other typically Sardinian horse breeds (Giara and Sarcidano) [7], although it lives in the same isolated geographic context. These results indicated the presence of multiple mare lineages in the extant SAA population, which were confirmed by the evidence of eight haplogroups out of the 18 identified by Achilli and colleagues [6] in the worldwide horse phylogeny. In particular, the frequency of the most ancient lineage, the haplogroup L (41%) identified in SAA, is strikingly higher than that recorded for the other three Italian breeds (17% in Bardigiano, 21% in Maremmano and 31% in Murgese) and for Continental Europe breeds (31%) [7]. This finding suggests a notable contribution of this lineage into the SAA female line from the Iberian Peninsula that was considered not only a place of refuge during the Last Glacial Maximum (∼25–19.5 kya) for many species but also one of the West-European areas where horse domestication occurred [6]. Even though we found a considerable number of shared haplotypes (mainly with Maremmano), the peculiarity of 18 SAA unique haplotypes needs to be highlighted and suggests the lack of crossbreeding and bottleneck phenomena along the maternal lines. Both mitochondrial haplotype and haplogroup composition agree with the indigenous origin of mares, which are distant from other Italian breeds and from Arabian and Thoroughbred lines.

## 5. Conclusions

The Sardinian Anglo-Arab horse represents one of the most important horse breeds in Italy, which has a significant historical background in Sardinia island. It originated from the crossbreeding between local mares and Arabian and Thoroughbred stallions, which strikingly affected its composition and demographic and genetic patterns. The comparison between the SAA population with three other Italian horse breeds (Bardigiano, Maremmano and Murgese) allowed us to define pedigree completeness, genetic variability and population structure of this Sardinian breed.

The estimated parameters on SAA suggest that the breed is well managed, especially considering the female lineage, since it still conserves a high number of founder mares. The pedigree completeness of living animals is close to 100%; this value is similar to that observed in the Maremmano breed that includes a smaller number of individuals.

The number of founders and ancestors is quite high, and their ratio shows that this population could be affected by a certain bottleneck effect and genetic drift, which are also confirmed by the effective numbers that are higher than those observed in other Italian horse breeds, thereby requiring special care to avoid these events. On the other hand, the average relationship coefficient is rather low, and the GCI is high in comparison with the values estimated on other Italian breeds; therefore, a good genetic variability can be presumed in SAA.

This finding could also be observed from a female perspective, wherein the mitochondrial analysis shows a high haplotype diversity, indicating the lack of bottleneck phenomena along the maternal lines and the important contribution of L lineage into the mtDNA gene pool, which is probably due to the re-colonization from the Iberian Peninsula after the Last Glacial Maximum.

## Figures and Tables

**Figure 1 animals-10-01018-f001:**
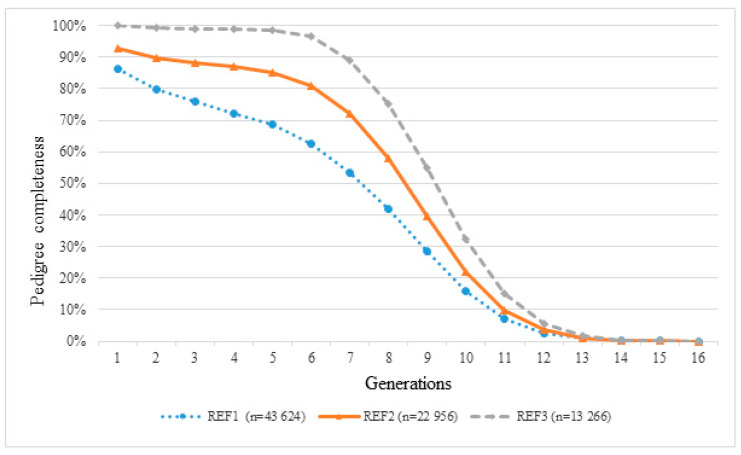
The completeness of pedigree information for the three Sardinian Anglo-Arab (SAA) reference populations.

**Figure 2 animals-10-01018-f002:**
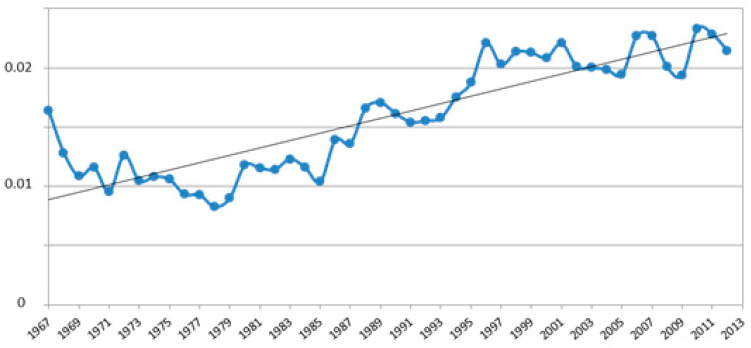
The average inbreeding coefficient (F) by year of birth in the REF1 population.

**Figure 3 animals-10-01018-f003:**
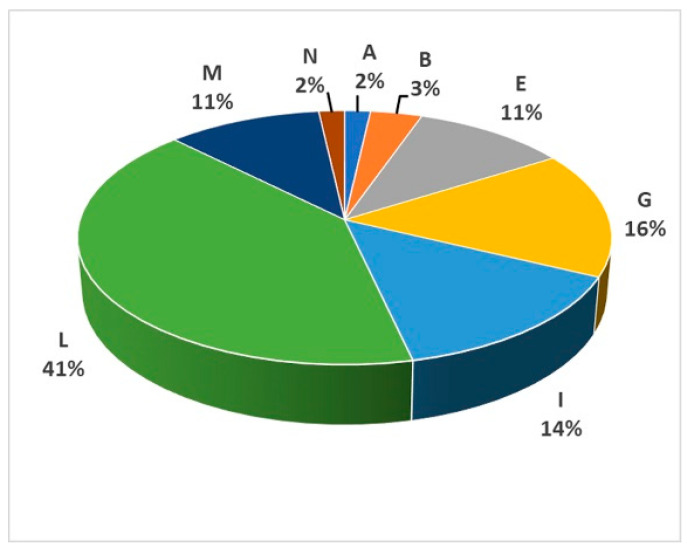
MtDNA haplogroup frequency distribution in the SAA.

**Figure 4 animals-10-01018-f004:**
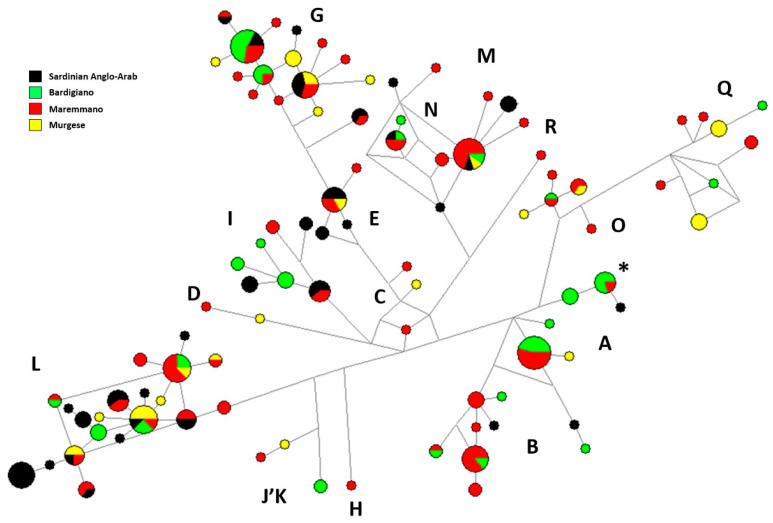
Median-joining network based on mitochondrial DNA (mtDNA) control-region sequences of four Italian horse breeds: Sardinian Anglo-Arab (black); Bardigiano (green); Maremmano (red); Murgese (yellow). Circles are proportional to the numbers of samples carrying the same haplotypes. The asterisk indicates the haplotype identical to ERS (Equine Reference Sequence; NC_001640.1). Capital letters refer to mtDNA haplogroups.

**Table 1 animals-10-01018-t001:** Sardinian Anglo-Arab reference populations compared with other Italian horse breeds.

Breed	Reference Population	*n*	Note
SAA	REF1	43,624	All the horses registered in the Studbook.
	REF2	22,956	Only Anglo-Arab, Sardinian Anglo-Arab, Thoroughbred, Arab and Sardinian Arab.
	REF3	13,266	Living horses.
MAREMMANO	REF1_MA	15,875	All the horses registered in the historical archive.
	REF2_MA	14,271	Horses after the exclusion of animals having Thoroughbred parents.
	REF3_MA	12,368	All the horses registered in the Studbook.
	REF4_MA	5705	Living horses.
BARDIGIANO	REF_B	9469	All the horses registered in the Studbook.
MURGESE	REF_M	2366	All the horses registered in the Studbook.

**Table 2 animals-10-01018-t002:** Pedigree completeness parameters of SAA, Maremmano, Murgese, Bardigiano.

Population	Maximum Number of Generations Traced	Fully Traced Generations	Equivalent Complete Generations
REF1	11.15	3.62	5.96
REF2	13.06	7.30	4.00
REF3	15.40	8.66	5.43
REF1_MA	-	-	-
REF2_MA	-	-	-
* REF3_MA	10.50	3.30	5.70
* REF4_MA	11.60	3.80	6.40
** REF_M	8.84	3.28	5.00
*** REF_B	-	-	-

* Giontella et al. [10], ** Bramante et al. [11], *** Ablondi et al. [9].

**Table 3 animals-10-01018-t003:** Genetic variability parameters of SAA, Maremmano, Murgese, Bardigiano.

Parameters	REF1	REF2	REF3	* REF3_MA	* REF4_MA	** REF_M	*** REF_B
F	0.012	0.017	0.021	0.029	0.036	0.045	0.080
AR	1.72	2.48	2.83	5.52	6.13	2.45	11.00
f_e_ (n)	287	190	157	74	64	37	18
f_a_ (n)	78	48	38	30	25	20	14
f_e_/f_a_	3.68	3.96	4.13	2.47	2.56	1.85	1.20
GCI	26.17	-	-	5.50	-	-	-
	±19.58	-	-	±3.37	-	-	-

* Giontella et al. [10], ** Bramante et al. [11], *** Ablondi et al. [9].

**Table 4 animals-10-01018-t004:** Populations structure parameters of SAA, Maremmano, Murgese, Bardigiano.

Parameters	REF1	REF2	REF3	REF1-MA	REF2-MA	* REF3-MA	* REF4_MA	** REF_M	*** REF_B
N_e_1_	456.86	196.32	187.02	-	-	68.10	71.20	-	-
N_e_2_	184.75	127.86	88.34	-	-	42.00	36.60	-	-
N_e_3_	209.31	128.52	87.36	-	-	42.20	36.90	-	-
N_e_1_/2	228.43	98.16	93.51	-	-	34.00	36.00	-	-
GI	10.98	-	-	-	-	10.65	-	11.70	8.74
	±5.10	-	-	-	-	±4.72	-	±10.90	-

* Giontella et al. [10], ** Bramante et al. [11], *** Ablondi et al. [9].

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
