# Peer review of "Genetic Variability and Population Structure in the Sardinian Anglo-Arab Horse"

_animals, 2020, doi:10.3390/ani10061018_

Round 1
Reviewer 1 Report
The authors of study entitled “Genetic variability and population structure in Sardinian Anglo-Arab horse” aimed to evaluate the genetic diversity of Sardinian Anglo-Arab (SAA) horses using pedigree and mitochondrial information. The authors provided a reviewed version of the manuscript addressing some of the comments. However, there are additional issues that must be clarified and/or fixed.
First of all, the paper must pass by an extensive English review regarding the grammar, the presence of typos and word choose. In several moments the reading is difficult (see below for a more detailed review).
Line 56: “These blood percentages [..]”. Is the word “these” referring to what? Were the authors intending to use “the blood percentages”?
Lines 59-60: This sentence is difficult to follow. Which breeds the authors would like to refer when wrote “one breed or the other”?
Line 62: “This study analyses” ïƒ This study analyzed.
Lines 64-65: “[…] could improve the next generation with their genes.”. This is a recurrent mistake across the manuscript. Disregarding events of deletion in the genome across generations, the same groups of genes are inherited. The differences between the animals are genetic variants mapped on these genes (or in intergenic regions) which could affect or not the functions performed by these genes.
Lines 62-65: I would recommend the authors to remove this paragraph due to the number of issues and also because the last paragraph introduce the aim of the study in a better way.
Line 67: replace “, such as” for “:”.
Line 95: Table 1. The authors should convert this table to a long format. The current version of this table is very bas formatted. The information would be better presented if the authors create a table with columns for population, reference, sample size and observations.
Line 106: Please, provide a brief explanation of the parameter used for this analysis.
Lines 106-108: Was the inbreeding coefficient normally distributed? Was the variance equal between the samples? This can affect completely the results of the t-test. Additionally, if the REF1 also contain the animals in REF3, would you expect independency between the samples? If not, the t-test is not the best option to compare the means of these two samples.
Line 119: Where is the start and the end of this sequence?
Line 137: “rose” ïƒ ??
Line 207: Table 2. Remove “Parameters”
Line 225: Table 3. Remove “Populations”
Line 238: “the highest value 0.08” replace for “the highest value was”
Line 288: “which have a good number of genes […]”. Again, the number of the genes in the genome is the same (disregarding events of duplication and deletion). The authors are trying to refer to the variants in the genome. Please, fix this conceptual mistake.
Line 296. Table 4. Remove “populations”
Line 315: These results really confirm that the population is not endangered or just show that the variability is higher than in other populations? In order to make this statement the authors must use thresholds that clearly state the conservation state of a population and/or specie.
Line 327: larger?
Lines 362-364: this paragraph is difficult to follow. Across the whole manuscript the authors are stating the population is not endangered, which is reinforced by the high number of ancestors and founders. What are these possible bottleneck events that happened in the past and how the authors reach the conclusion of the existence of these events?
Author Response
Please see the attachment.
The authors gratefully thank to the Referee for the constructive comments and recommendations which definitely help to improve the readability and quality of the paper. All the comments are addressed accordingly and have been incorporated to the revised manuscript. Detailed responses to the comments and recommendations are in the attach file .

Reviewer 2 Report
Thank you for addressing my comments, and congratulations on the manuscript. I recommend it for publication in its current form.
Author Response
Thanks. Best regards
Reviewer 3 Report
The authors analyzed pedigreed records and mtDNA sequences of the Sardinian Anglo-Arab horse to evaluate parameters of breed structure and genetic diversity that can be used for considerations of breed management. The methods used were conventional and appropriate, the conclusions were mostly aligned with the results, and relevant literature was referenced. The study has value to local horse breeding industry and more broadly to scientists who work in similar field.
Overall, the manuscript can benefit for further editing for language and fine-tuning of content. It tends to be unnecessarily verbose in places and it could be made more concise relative to the content. For example, this is much repetition in text of results that are already presented in tables. Much of this repetition can be avoided by description of general trends. A few additional suggestions for edits and clarification are:
Line 28: Replace “breeds” with “breed”.
Line 31: This should be qualified. Completeness is close to 100% for current population (living animals), according to results.
Line 37: replace “populations” with “horses”.
Lines 39-40: Insert “number” before “founder”. Delete text “the marginal impact of crossbreeding on the indigenous maternal gene pools” if this is based on material from Discussion. See comments below (lines 347-349, 370).
Line 57: Can the authors summarize what constitutes an SAA according to studbook requirements relative to genetic content of other breeds?
Lines 59-61: Move paragraph to end of line 48. Delete “As a matter of fact”, not needed.
Line 63: Delete “aforementioned”, not needed. Also not needed in several other places throughout manuscript.
Lines 78-79: Clarify the composition of reference populations and the rationale for each grouping. It seems that REF2 contains SAA and ancestors with Arabian and Thoroughbred content. What is then in the much larger REF1?
Lines 106-107: Revise sentence to “…a further analysis was performed to verify…”
Line 117: mtDNA variation: provide description of PCR conditions as the detail can be of interest to other researchers to use the same method in other studies.
Line 123: Revise: … and Sanger sequencing with the same primers.
Line 128: Replace “described” with “shown” or “presented”..
Lines 131-134: Description of results needs to be improved. “first to fifth generations” means ancestors of the breed. The plot represents the number of ancestral generations that can be traced in records.
Line 144: Legend of Figure 2: insert “by year of birth” after “(F)”.
Line 179: Insert “SAA” before haplotypes.
Line 241-242: What is the meaning of “consistency”? Is the intent to state that estimated F-values can be explained by differences in population size of registered horses in each breed? Please be more specific to meaning.
Lines 206-219 and 225-235: Tables 2 and 3 belong in the respective Results section.
Line 278: Can “massive use of same stallions” be quantified? Some examples would help readers understand the context of the “popular sire effect” in the breed.
Line 329: Use “sporting” instead of “sportive”.
Line 347-349: Crossbreeding with stallions from other breeds has no effect on maternal lineages. Perhaps what the authors mean is that the mtDNA results suggest that, likely, much of the genetic content and diversity contributed by the indigenous mares remains in the SAA, despite crossbreeding history.
Line 370: This statement “the marginal impact of crossbreeding on the indigenous mitochondrial gene pools” doesn’t make sense given the assertions that only stallions from other breeds contribute to SAA. It should be deleted or statement clarified.
Author Response
Please see the attachment.
The authors gratefully thank to the Referee for the constructive comments and recommendations which definitely help to improve the readability and quality of the paper. All the comments are addressed accordingly and have been incorporated to the revised manuscript. Detailed responses to the comments and recommendations are in the attach file .

This manuscript is a resubmission of an earlier submission. The following is a list of the peer review reports and author responses from that submission.
Round 1
Reviewer 1 Report
The authors of study entitled “Genetic variability and population structure in Sardinian Anglo-Arab horse” aimed to evaluate the genetic diversity of Sardinian Anglo-Arab (SAA) horses using pedigree and mitochondrial information. The paper is well written in general. However, there are several points that must be better explained, mainly regarding the discussion of the results.
Major comments:
The introduction is mainly focused on the history of SAA horses. It is important to contextualize the breed to the authors. However, the authors also should use the introduction to better explain the necessity and the which literature gaps need to be filled by the analysis performed in the manuscript. In summary, why study the genetic diversity of SAA?
The mitochondrial DNA variation subsection in the Material and Methods section must be better explained. The authors are presenting on lines 169-170 (Results section) all the information regarding the sample size. This information must be present in the material and method section. Additionally, which primers were used for the sequencing? Which was the PCR reaction program? Additionally, the authors also must better describe the evolutionary relationship analysis performed in this step.
The pedigree completeness subsection in the Discussion section is a summary of results. The authors must discuss the observed results regarding the scientific evidences that reinforce the observations. The authors must try to find evidences of historical events that help to explain the dynamic of the pedigree completeness across the generations. The same applies for all the other subsections. The authors are just presenting the results of the comparisons between the reference populations. This is highlighted by the fact that there is not a single new reference added in this section. What are the evolutionary events explaining the results? What are the consequences of the observed results? The results need to be discussed. In this section, the authors must fill the gaps present in the literature and bring contributions for the current status of the population genetic studies regarding horse populations. This is the main limitation of the study. The authors fail to discuss the observed results and the manuscript seems to be a simple description of the population diversity of a regional breed.
The authors close the conclusion section with the following sentence: “Furthermore, this evidence confirms the potentiality of the mitochondrial DNA as a powerful molecular marker in genetic studies, pedigree validation and genetic conservation programs.” There are decades of studies using mitochondrial DNA for population genetic purposes in other species. The advantages and limitations (several limitations, indeed) are very well described in the literature. This information should be present in the discussion section and there is not novelty in this sentence. Again, reinforcing the main limitation of the study.
Minor comments:
Line 31: “the average relatedness (<3%) were lower” compared to what?
Line 33: “smaller breeds”. Are the authors referring to the height of the animals or to the population size? If the comment was regarding the population size, I would suggest the authors to re-write the sentence.
Lines 33-34: “that are always greater than”. Which scenarios were compared?
Lines 42-49: A reference must be provided for this information, if possible.
Line 66: this is a typical sentence from introduction or discussion, I suggest removing it from the material and methods section.
Table 1: The format of this table is very difficult to follow. I suggest the author to reformat the table. Additionally, probably due to the temporary status of the pdf created for the review processes, the table is split in a way that the information from REF1_MA is truncated.
Lines 88-89: Were the criteria used for the pedigree analysis or all reported parameters related to pedigree completeness previous reported? Checking the cited reference, it seems that the authors would like to mention the pipeline used to evaluate the pedigree parameters. This is not clear in the sentence. Additionally, I would suggest the authors to include this criteria even if informed the previous paper.
Reviewer 2 Report
In this paper, the authors describe the genetic diversity, population structure and mitochondrial DNA haplotype structure of the Sardinian Anglo-Arab horse, and compare these parameters with those of three other small Italian horse breeds: Maremmano, Murgese and Bardigiano. Results reveal that the SAA breed is more genetically diverse, has a higher effective number of individuals, and a more diverse collection of mitochondrial haplotypes when compared to the other breeds used in this study. The authors attribute these findings to a better breed management program.
This study is interesting and original. The work is well done, and results improve our knowledge on genetic diversity and relationships among Italian horse breeds.
I have some suggestions for the presentation of the study:
L.17 – remove “the”; Pedigree does not need to be capitalized
L.19 – remove “others”
L.19 – replace with “three other small Italian horse populations” for clarity
L.20 – I suggest replacing “origin” with “inheritance” or “lineage”
L.22-23 – replace “mare founders” with “founder mares”
L.27 – I suggest replacing “good mating programs” with “effective mating programs”
L.28 – Reference Populations does not need to be capitalized
L.30 – Remove “the” from “The pedigree completeness”
L.31 – Remove “the” from “the inbreeding rate” as well as from “the average relatedness”
L.31 – replace “inbreeding rate” with “inbreeding coefficient”
L.32-33 – “The ratio of effective founders / numbers of ancestors for SAA was higher than that computed on smaller breeds, indicating a genetic drift and bottleneck phenomena.” – confusing, please clarify
L.33 – the results of the paper do not indicate genetic drift or a bottleneck phenomenon, please clarify.
L.34 – Replace “numbers” with “number”
L.35 – remove “the” from “the mitochondrial DNA”
L.41 (Introduction) – the introduction can benefit from contextualizing what has been described in terms of genetics of the SAA breed in relation to other small Italian populations used in this study.
L.57 – I would suggest replacing “ethnological” with “composition” or “breed composition”
L.60-63 – This sentence is long and thus confusing. I would suggest to break it up into 2-3 smaller sentences.
L.66-67 - I would suggest removing this sentence.
L.68-69 - Replace with “In this study, three reference populations (REF) were considered: population REF1 consists…”
L.70 – remove “the” from “the REF2”
L.70 – add “horses” after “REF2 consists of 22,956”
L.72 – remove “the” from “the REF3”.
L.72 – REF3 includes living horses registered in the A.N.A.C.A.A.D (subset of REF1)? Please clarify.
L.83-87 – a brief explanation on how these parameters are calculated is needed.
L.85 – Replace “individual inbreeding-F” with “average inbreeding coefficient”
L.87 – ENDOG software, not Endog.
L.88 – Replace “overcome a missing interpretation of” with “avoid a possible misinterpretation of the”
L.88-91 - a brief explanation on how these parameters are calculated is needed.
L.89 – Replace “small consistency of population” with “small number of individuals”
L.95-97. a brief explanation on how these parameters are calculated is needed. How many regions? How were they analyzed and compared?
L.97 – Remove “here considered”
L.99 – I would suggest replacing “the tree construction performed by Network software” with “a phylogenetic tree constructed using the “Network”v.XX software”
L.99 – Please indicate which version of the Network software was used (see above).
L.100 (Results section as a whole): statistical analyses need to be performed for any of the results to be meaningful, since this is a comparison between populations.
L.102 – I would suggest replacing “set out” with “described”
L.104 – This figure legend is incomplete and needs to be expanded
L.105-108 – this description is redundant because it can be obtained from the figure, and can be removed.
L.110 – The table caption is incomplete and needs to be expanded.
L.111-119(Table 2) – Data on other reference populations missing from the table: REF1_MA, REF2_MA, REF_B. The table also needs to be formatted correctly, to follow the text referring to it above (Materials and Methods). Please indicate values that are not available for populations with N/A or - instead of leaving blank spaces.
L.119 – references should be denoted with their respective numbers for consistency.
L.120-123 –Again, this explanation is redundant and can be removed.
L.125 – “Reference populations” does not need to be capitalized
L.129 – F is reported in percentages here, please represent F values in absolute values (0.012, etc)
L.136 – references should be denoted with their respective numbers for consistency.
L.137-138 – Is this difference statistically significant? If a statistical analysis is not performed on these data, reporting them is not useful. Also, why is it only reported for REF1 and not the other reference populations?
L.141-149 - Again, all this information can be obtained from the table, so this should be removed. Individual values described herein can be described in supplementary tables.
L.152 - The table caption is incomplete and needs to be expanded.
L.153-160 (Table 4) – this table needs to be re-formatted. Some values have 2 decimals whereas some have only one. Please indicate values that are not available for populations with N/A or - instead of leaving blank spaces.
L.162-167. Again, all this information can be obtained from the table, so this should be removed. Individual values described herein and used to calculate averages can be described in supplementary tables.
L.178 (Figure 3) – Please indicate the frequency of each haplogroup in the pie chart
L.180 – Replace “The network analysis” with “Network analysis”
L.182 – Replace “haplogroups clustering” with “haplogroup clustering”
L.184-186 – Color-coding and letters in the figure need to be explained in the figure legend.
L.185 – Remove “here considered”.
L.194-249. This discussion is mostly descriptive, and repeats the data described in the results section. Instead, the authors should focus on providing biological and evolutionary context to their findings in SAA in relation to other Italian breeds. Extensive re-writing of sections 4.1 to 4.3 is needed.
L.264 – Replace “mare founders” with “founder mares”
L.266 – I suggest Replacing “heads” with “individuals”
L.267-270- How did the authors reach this conclusion? Please clarify.
L.272 – this conclusion cannot be reached without performing statistical analyses to determine if the differences in genetic variability between breeds are significant.
Reviewer 3 Report
I do not understand why this breed is only compared with other Italian breeds. I would assume to include its founders (Arabian horses and Thoroughbred). How many maternal linages do exists in the founder breeds? How many of them were introgressed etc.? I am missing also any genetic test of pedigree data. Its a well known fact that many studbooks are full of errors. How reliable are these data? Finally, a breed is an arbitrary unit which exists for a certain number of years/decades. Inbreeding in a domestic breed has always to be discussed in relation to the entire domestic species because it can easily to be reduced by introgression of stallions from other breeds. Therefore, inbreeding is less important in domestic species then in natural populations. Inbreeding is a powerful tool for breed improvement and fixation of traits - an issue which is completely missing in this article.
Checking list of references, I would strongly suggest to merge this article with Giontella A., Sarti F. M., Biggio G. P., Giovannini S., Cherchi R., Silvestrelli M., Pieramati C.; 291 Genetic parameters and inbreeding effect of morphological traits in Sardinian Anglo Arab horse. 292 Animals, under review.
Round 2
Reviewer 2 Report
Response to Reviewer 4 Comments
In this paper, the authors describe the genetic diversity, population structure and mitochondrial DNA haplotype structure of the Sardinian Anglo-Arab horse, and compare these parameters with those of three other small Italian horse breeds: Maremmano, Murgese and Bardigiano. Results reveal that the SAA breed is more genetically diverse, has a higher effective number of individuals, and a more diverse collection of mitochondrial haplotypes when compared to the other breeds used in this study, attributing these to a better breed management program.
This study is interesting and original. The work is well done, and results improve our knowledge on genetic diversity and relationships among Italian horse breeds.
I have some suggestions for the presentation of the study:
L.17 – remove “the”; Pedigree does not need to be capitalized
Response: Accepted and amended.
We remove “The”
Reviewer 4 response: Thank you.
L.19 – remove “others”
Response: Accepted and amended.
We remove “others
Reviewer 4 response: Thank you.
L.19 – replace with “three other small Italian horse populations” for clarity
Response: Accepted and amended.
We accepted your suggestion and we changed the sentence with: “three other small Italian horse populations”
Reviewer 4 response: Thank you.
L.20 – I suggest replacing “origin” with “inheritance” or “lineage”
Response: Accepted and amended.
We removed “origin” with “lineage”
Reviewer 4 response: Thank you.
L.22-23 – replace “mare founders” with “founder mares”
Response: Accepted and amended.
We changed “mare founders” with “founder mares”
Reviewer 4 response: Thank you.
L.27 – I suggest replacing “good mating programs” with “effective mating programs”
Response: Accepted and amended.
We changed “good mating programs” with “effective mating programs”
Reviewer 4 response: Thank you.
L.28 – Reference Populations does not need to be capitalized
Response: Accepted and amended.
We removed the capital letters.
Reviewer 4 response: Thank you.
L.30 – Remove “the” from “The pedigree completeness”
Response: Accepted and amended.
We removed “the”.
Reviewer 4 response: Thank you.
L.31 – Remove “the” from “the inbreeding rate” as well as from “the average relatedness”
Response: Accepted and amended.
We removed “the” from “the inbreeding rate” as well as from “the average relatedness”.
Reviewer 4 response: Thank you
L.31 – replace “inbreeding rate” with “inbreeding coefficient”
Response: Accepted and amended.
We replaced “inbreeding rate” with “inbreeding coefficient”.
Reviewer 4 response: Thank you.
L.32-33 – “The ratio of effective founders / numbers of ancestors for SAA was higher than that computed on smaller breeds, indicating a genetic drift and bottleneck phenomena.” – confusing, please clarify
Response: Accepted and amended.
We changed the sentence as following: “The ratio of effective founders / numbers of ancestors for SAA was higher than that computed on populations with smaller consistency, indicating a genetic drift and bottleneck phenomena”.
Reviewer 4 response: Please replace “populations with smaller consistency” with “populations with a lower number of individuals”.
L.33 – the results of the paper do not indicate genetic drift or a bottleneck phenomenon, please clarify.
Response: Accepted and amended.
As suggested from another Referee we changed in the discussion at line 317 the sentence referred to the bottleneck phenomena in SAA. Please verify if this form is clearer.
Reviewer 4 response: Line 34 – replace “effective population number” with “effective population size”; Ne – the “e” should be subscript.
L.34 – Replace “numbers” with “number”
Response: Accepted and amended.
We replaced “numbers” with “number”.
Reviewer 4 response: Thank you.
L.35 – remove “the” from “the mitochondrial DNA”
Response: Accepted and amended.
We removed “the” from “the mitochondrial DNA”.
Reviewer 4 response: Thank you.
L.41 (Introduction) – the introduction can benefit from contextualizing what has been described in terms of genetics of the SAA breed in relation to other small Italian populations used in this study.
Response: Accepted and amended.
We enriched the introduction at line 77 with the follow sentence: “SAA reference populations were analysed and compared with three other small Italian horse populations, analysed with the same statistical approach, such as Bardigiano, Maremmano and Murgese”.
Reviewer 4 response: Thank you.
L.57 – I would suggest replacing “ethnological” with “composition” or “breed composition”
Response: Accepted and amended.
We changed “ethnological”with “breed composition”
Reviewer 4 response: Thank you.
L.60-63 – This sentence is long and thus confusing. I would suggest to break it up into 2-3 smaller sentences.
Response: Accepted and amended.
We break up the line 60-63 in two sentences.
Reviewer 4 response: Thank you.
L.66-67 - I would suggest removing this sentence.
Response: Accepted and amended.
We moved the sentence in the introduction as suggested by another Referee.
Reviewer 4 response: Thank you.
L.68-69 - Replace with “In this study, three reference populations (REF) were considered: population REF1 consists…”
Response: Accepted and amended.
We modified the sentence as suggested.
Reviewer 4 response: Thank you.
L.70 – remove “the” from “the REF2”
L.72 – remove “the” from “the REF3”.
Response: Accepted and amended.
We remove “the” from “the REF2” and “the REF3”
Reviewer 4 response: Thank you.
L.70 – add “horses” after “REF2 consists of 22,956”
Response: Accepted and amended.
We added “horse” as suggested.
Reviewer 4 response: Thank you.
L.72 – REF3 includes living horses registered in the A.N.A.C.A.A.D (subset of REF1)? Please clarify.
Response: Accepted and amended.
REF 3 is referred only to living horses, both REF 2 and REF 3 are subset of the REF1 the total A.N.A.C.A.A.D Studbook.
Reviewer 4 response: Thank you.
L.83-87 – a brief explanation on how these parameters are calculated is needed.
The parameters were calculated by ENDOG software, we cited it with another article that explain the same method (Reference 11). While the article was made from the same authors, we thought that write it again could have been a plagiarism. If the Referee believes that more explanations are necessary we will add that.
Reviewer 4 response: The reference (11) will suffice, thank you.
L.85 – Replace “individual inbreeding-F” with “average inbreeding coefficient”
“individual inbreeding-F” is a parameter computed by ENDOG, and it is called in this way in the official manual.
Reviewer 4 response: Thank you.
L.87 – ENDOG software, not Endog.
Response: Accepted and amended.
We replaced “Endog” with ENDOG.
Reviewer 4 response: Thank you.
L.88 – Replace “overcome a missing interpretation of” with “avoid a possible misinterpretation of the”
Response: Accepted and amended.
We replaced “overcome a missing interpretation of” with “avoid a possible misinterpretation of the”.
Reviewer 4 response: Thank you.
L.88-91 - a brief explanation on how these parameters are calculated is needed.
We added reference that contains more explanation. We choose not to explain how the parameters are computed because it is reported in the manual of ENDOG, which is reported as a reference.
Reviewer 4 response: Line 96-99 – please re-write these sentences since they are syntactically incorrect.
L.89 – Replace “small consistency of population” with “small number of individuals”
Response: Accepted and amended.
We replaced “small consistency of population” with “small number of individuals”.
Reviewer 4 response: Thank you.
L.95-97. a brief explanation on how these parameters are calculated is needed. How many regions? How were they analyzed and compared?
Response: Accepted and amended.
We added the suggested information.
Reviewer 4 response: Thank you.
L.97 – Remove “here considered”
Response: Accepted and amended.
We removed it.
Reviewer 4 response: Thank you.
L.99 – I would suggest replacing “the tree construction performed by Network software” with “a phylogenetic tree constructed using the “Network”v.XX software”
Response: We replaced with “through the median-joining phylogenetic tree construction performed by Network software 10.0 (www.fluxusengineering.com)”
Reviewer 4 response: Thank you. Line 111- check for errors.
L.99 – Please indicate which version of the Network software was used (see above).
Response: Accepted and amended.
We indicated the Network software version in the sentence above reported.
Reviewer 4 response: Thank you.
L.100 (Results section as a whole): statistical analyses need to be performed for any of the results to be meaningful, since this is a comparison between populations.
Response: The present study is performed only on SAA horse population, so this section is entirely dedicated on SAA statistical analyses. Other authors that we reported in References have performed the statistical analysys of the other Italian horse population. In Discussion, we mentioned the results of these studies.
L.102 – I would suggest replacing “set out” with “described”
Response: Accepted and amended.
We replaced “set out” with “described”.
Reviewer 4 response: Thank you.
L.104 – This figure legend is incomplete and needs to be expanded
Response: Accepted and amended.
We implemented the figure legend as following: “The completeness of pedigree information for the three SAA reference populations”
Reviewer 4 response: Thank you.
L.105-108 – this description is redundant because it can be obtained from the figure, and can be removed.
Response:We thank the Referee for the suggestion, but we believe that it is difficult to reach the values reported in lines 105-108 by only the figure. In the figure we choose entire numbers, while in the explanation we reported the specific values that are not entire.
Reviewer 4 response: Thank you.
L.110 – The table caption is incomplete and needs to be expanded.
Response: Accepted and amended.
We replaced the caption in Table 2 with the complete names of Pedigree completeness parameters.
Reviewer 4 response: Thank you.
L.111-119(Table 2) – Data on other reference populations missing from the table: REF1_MA, REF2_MA, REF_B. The table also needs to be formatted correctly, to follow the text referring to it above (Materials and Methods). Please indicate values that are not available for populations with N/A or - instead of leaving blank spaces.
Response: Accepted and amended.
We added the other populations. We decided, after the Referee’s suggestion, to move the complete Table (with the other populations values added) in the Discussion section of this article.
Reviewer 4 response: Thank you. But why move the table to the Discussion? I would keep it in the Results section.
L.119 – references should be denoted with their respective numbers for consistency.
Response: Accepted and amended.
We corrected the references as following: “*Giontella et al., 2019[11]**Bramante et al., 2017[12]“
Reviewer 4 response: Thank you. What I meant was *[11], **[12]
L.120-123 –Again, this explanation is redundant and can be removed.
Response: Since the information written plus the information in the Table are redundant together, we decided to move the Table in the Discussion section.
Reviewer 4 response: Thank you. But why move the table to the Discussion? I would keep it in the Results section.
L.125 – “Reference populations” does not need to be capitalized
Response: Accepted and amended.
We removed the capital letter in “Reference populations”.
Reviewer 4 response: Thank you.
L.129 – F is reported in percentages here, please represent F values in absolute values (0.012, etc)
Response: Accepted and amended.
Reviewer 4 response: Thank you. But why move the table to the Discussion? I would keep it in the Results section.
L.136 – references should be denoted with their respective numbers for consistency.
Response: Accepted and amended.
Reviewer 4 response: Thank you. What I meant was *[11], **[12], ***[10]
L.137-138 – Is this difference statistically significant? If a statistical analysis is not performed on these data, reporting them is not useful. Also, why is it only reported for REF1 and not the other reference populations?
Response: We decided to represent in the Figure 2 the trend of the entire population, which is described only by REF1.
We computed the F also for the other reference populations (only SAA), as reported at line 327.
Reviewer 4 response: What I’m referring to is – does the increase in F from 0.012 to ~0.02 since 1967 mean that the population is getting more inbred over time? If 0.02 is statistically significantly higher than 0.012, this means yes. This information is of extreme importance for breeders and owners.
L.141-149 - Again, all this information can be obtained from the table, so this should be removed. Individual values described herein can be described in supplementary tables.
The description of the Table 4 at line 348 is focused only on SAA reference populations, but in the Table we reported also the other breeds parameters, so we decided to move the Table in the Discussion section. In this way we avoid the redundant information in the Results.
Reviewer 4 response: Thank you. But results should be kept in the Results section. I would not move the table to the Discussion section. Table caption is still incomplete, it says “in the three SAA reference populations” but includes data on other populations as well.
L.152 - The table caption is incomplete and needs to be expanded.
Response: Accepted and amended.
Table capitation are now complete.
Reviewer 4 response: Table caption is still incomplete, it says “in the three SAA reference populations” but includes data on other populations as well.
L.153-160 (Table 4) – this table needs to be re-formatted. Some values have 2 decimals whereas some have only one. Please indicate values that are not available for populations with N/A or - instead of leaving blank spaces.
Response: Accepted and amended.
Table 4 has been implemented with the information suggested.
Reviewer 4 response: Table caption is still incomplete, it says “in the three SAA reference populations” but includes data on other populations as well.
L.162-167. Again, all this information can be obtained from the table, so this should be removed. Individual values described herein and used to calculate averages can be described in supplementary tables.
Response: Accepted and amended.
We decided, after the Referee’s suggestion, to move the complete Table 4 (with the implementation suggested) in the Discussion section of this article (4.3. Population structure).
We believe that in this way it might be less redundant and clearer for readers.
L.178 (Figure 3) – Please indicate the frequency of each haplogroup in the pie chart
Response: Accepted and amended.
We indicated the frequency of each haplogroup in the pie chart.
Reviewer 4 response: Thank you.
L.180 – Replace “The network analysis” with “Network analysis”
Response: Accepted and amended.
We replaced “The network analysis” with “Network analysis”.
Reviewer 4 response: Thank you.
L.182 – Replace “haplogroups clustering” with “haplogroup clustering”
Response: Accepted and amended.
We replaced “haplogroups clustering” with “haplogroup clustering”.
Reviewer 4 response: Thank you.
L.184-186 – Color-coding and letters in the figure need to be explained in the figure legend.
We modified the legend as following: “Median-Joining Network based on control-region sequences of four Italian horse breeds: Sardinian Anglo-Arab (black); Bardigiano (green); Maremmano (red); Murgese (yellow). Nodes are proportional to the number of samples carrying the same haplotype. The asterisk indicates the haplotype identical to ERS (Equine Reference Sequence; NC_001640.1). Capital letters refer to haplogroups.”.
Reviewer 4 response: Thank you.
L.185 – Remove “here considered”.
Response: Accepted and amended.
We removed “here considered”.
Reviewer 4 response: Thank you.
L.194-249. This discussion is mostly descriptive, and repeats the data described in the results section. Instead, the authors should focus on providing biological and evolutionary context to their findings in SAA in relation to other Italian breeds. Extensive re-writing of sections 4.1 to 4.3 is needed.
Response: Accepted and amended.
4.1 has been implemented.
4.2 has been implemented.
4.3 has been implemented.
Reviewer 4 response: Thank you, the discussion sections have improved in content.
L.264 – Replace “mare founders” with “founder mares”
Response: Accepted and amended.
We replace “mare founders” with “founder mares”.
Reviewer 4 response: Thank you.
L.266 – I suggest Replacing “heads” with “individuals”
Response: Accepted and amended.
We replace “heads” with “individuals”.
Reviewer 4 response: Thank you.
L.267-270- How did the authors reach this conclusion? Please clarify.
Response: Accepted and amended.
We implemented the discussion. At lines 256-259 and 443-447 we clarified why we declare that in SAA population bottleneck phenomena may occur.
Reviewer 4 response: Thank you. It is not clear to me how that might lead to a population bottleneck. It might certainly lead to a decrease in genetic diversity (as your increasing F values might indicate if significant), but how can it lead to a population bottleneck?
L.272 – this conclusion cannot be reached without performing statistical analyses to determine if the differences in genetic variability between breeds are significant.
Response: We thank the referee for the suggestion, but we can’t applied this approach to our data for two reasons:
- we declare that we are not in possess of Murgese and Bardigiano pedigree, so we cannot perform statistical analyses;
- the major software used for the study, structure, inbreeding and genetic variability of a population from a pedigree analysis is ENDOG. ENDOG (current version v4.8: a computer program for monitoring genetic variability of populations using pedigree information. Journal of animal breeding and genetics, 122:172-176) is a population genetics computer program that conducts several demographic and genetic analyses on pedigree information in a user friendly environment. The program help researchers or those responsible for management of populations to monitor the changes in genetic variability and population structure with a limited amount of prior preparation of datasets. Although it is used primarily as a population-monitoring package, ENDOG does not provide any statistical analysis about significant difference in genetic variability between breeds.
Reviewer 4 response: Thank you. I caution the authors about the conclusions reached from the comparisons with other horse populations without a means to scientifically demonstrate such conclusions. Therefore, I am weary to consider the validity of this study, especially when it comes to the assumptions laid out by the authors in the Discussion.
Addendum: please revise the newly written portions of the manuscript, as they contain numerous syntax errors.

Reviewer 3 Report
The manuscript has not been improved. However, it’s an editorial decision if or if not be accepted. In my opinion it is a case of salami slicing which I would not support.